# Entrepreneurial Behavior of SMEs and Characteristics of the Managers of Northwest Mexico

**Sergio Ochoa Jiménez *** , **Alma Rocío García García, Beatriz Alicia Leyva Osuna and Sacnicté Valdez del Río**

Department of Administrative Sciences, Instituto Tecnológico de Sonora, Cajeme 85000, Sonora, Mexico; alma.garcia@itson.edu.mx (A.R.G.G.); beatriz.leyva@itson.edu.mx (B.A.L.O.); sacnicte.valdez@itson.edu.mx (S.V.d.R.)
* Correspondence: sochoa@itson.edu.mx

**Abstract:** The objective of this study is to determine the difference in the entrepreneurial behavior of companies based on the demographic characteristics of their manager or leader. To comply with the above, a quantitative, transversal, and non-experimental research study was carried out, which consisted in applying an instrument to 262 managers of small and medium enterprises (SMEs) in a northern city in Mexico. The collected information was analyzed in the software SPSS, version 26, with statistical testing by the Mann-Whitney and Kruskal-Wallis tests. The main findings show that these companies have differences in their entrepreneurial behavior based on the age and educational level of their managers, while gender and seniority at work are not differentiating elements in relation to the above. This research generates different possibilities of studies to be carried out in large companies from other sectors, and suggests the inclusion of behavioral characteristics as study variables.

**Keywords:** management; entrepreneurial behavior; leadership; SMEs; innovation

## 1. Introduction

Over the years, the importance of SMEs has lain in the role they have played as a driving force in employment and in their contribution to better development and economic growth in many countries; therefore, it is advisable to continue promoting their creation in various sectors (Palomo González 2005). In addition to generating jobs and boosting the economy, they are also able to adapt to different aspects, whether technological or social, thus achieving better performance (Delgado Delgado and Chávez Granizo 2018).

Few SMEs are successful and manage to stay in the market for years. Some of them become large, competitive, and productive companies (Saavedra García et al. 2018; Atristain and Rajagopal 2010) and generate significant economic income (Jiménez Martínez 2007; Gonzales Herrera 2011; Chong et al. 2019). Entrepreneurs focus on achieving the legal requirements that regulate SMEs in order to preserve profits, but at the same time limit their growth (Tsuruta 2020). Along with the above, SMEs have other impediments to their competitiveness and growth, such as lack of capital and skills (Maksum et al. 2020), basic and traditional use of technology (Mohd Selamat et al. 2020), as well as others. For this reason, these companies have been studied from different perspectives and approaches, and by numerous countries.

### 1.1. SMEs and Their Areas of Study

Among the general studied themes related to the SMEs, there is the strategy of differentiating elements in the companies and focusing on the development of their planning (Maldonado and Erazo 2015; Sánchez 2003; Lozano 2010; Velásquez Vásquez 2004). In addition, it is important that companies ensure their competitiveness to improve their performance and achieve competitive success as an essential part of their promising future (Antonio and Bañón 2005; Romero and Santoyo 2009; Cano Flores et al. 2014; Saoudi and

Foliard 2019; Martínez Rojas et al. 2013). The importance of these entrepreneurial companies in the economy of their countries and the generation of cross-border transactions (Lee et al. 2020) was previously mentioned; therefore, the topic of entrepreneurship has been analyzed in the SMEs (Sánchez et al. 2014; Cruz and Justo 2017; Sciascia et al. 2006). It is presented due to the growth and helps to fulfill proposed objectives, as well as to focus not only on economic aspects, but also on emotional ones (Cruz and Justo 2017); additionally, it supports knowledge acquisition, market orientation and commitment, and with it to continue growing and developing in the market (Sciascia et al. 2006).

For improvement to occur in an SME, it is necessary to know its business orientation and objectives, and thus implement strategies by examining their effects (Felzensztein et al. 2015) with a positive implementation directed towards performance (Butkouskaya et al. 2020). To achieve this, it is important to have learning perspectives at all organizational levels (Brettel and Rottenberger 2013; Altinay et al. 2016); to be clear about the importance of an external orientation that seeks to drive the benefits of organizations (Brettel et al. 2015); the ease of creating knowledge and sharing it in order to increase their competitiveness (Pérez-Luño et al. 2016); and to recognize the fundamental role of knowledge in the external market. Related to business orientation in SMEs, those companies focused on innovation with a strong organizational culture, concerned about a good development and growth in the future (Basile 2012), are highlighted, since an innovative business model has a positive and significant impact on the competitiveness and performance of SMEs (Anwar 2018).

Sustainability has been studied as an emerging issue in SMEs (Bartolacci et al. 2020; Bakos et al. 2020; Jansson et al. 2017), as the topic has aroused great interest not only in academia and society, but also at the business level (Bartolacci et al. 2020). One of the challenges they face is climate change, so they should always consider environmentally friendly and sustainable practices (Bakos et al. 2020), with the fact that it has become a very relevant factor within the business environment (Jansson et al. 2017). Another addressed topic is human resource management, starting with all the members of the organization who contribute to achieving the proposed goals and objectives; it is also considered a determining key in guiding the course of the organization (Harney and Alkhalaf 2020) and focusing on needs, such as training to achieve better control and good performance within the organization (Bermúdez Carrillo 2015).

Another addressed topic is human resource management, starting with all the members of the organization who contribute to achieving the proposed goals and objectives; it is also considered a determinant in guiding the course of the organization (Harney and Alkhalaf 2020) and focusing on needs, such as training to achieve better control, and good performance within the organization (Bermúdez Carrillo 2015).

Consequently, in order for SMEs to transcend, it is necessary to focus on their finances and the existing accounting of their organizational structure, since they are considered a source of competitive advantage and an essential element for making decisions (García Pérez De Lema et al. 2006); nevertheless, the main challenge they face is the competitive environment, since dealing with it requires effort. That is why there are several systems that allow SMEs to be properly managed as accounting systems that bring benefits in terms of performance (López Mejía and Hernández 2010) and thus are able to reduce risks (Berger and Schaeck 2011). Those that are related to the operation of the companies and their supply chain must be identified in order to generate proposals to mitigate the impact (Fan and Stevenson 2018). Internally, insufficient utility may be of greater risk than other situations (Oláh et al. 2019); externally, it is important to locate economic, geopolitical, social, technological, and environmental risks, of which the latter are considered to be of lesser impact (Asgary et al. 2020).

For SMEs, their performance is fundamental, so being interested in planning and control factors has become imperative for companies to plan carefully and thereby reduce difficulties (Yusuf and Saffu 2005), due to the fact that in some cases they show structures that are not well planned, preventing them from increasing their performance (Cortés et al.

2016). That is why it is important to design strategies that bring positive effects to the organization (Radicic and Pugh 2017), as well as structured work systems, proper resource management, and knowledge creation, among others (Klaas et al. 2006).

### 1.2. The Entrepreneurial Behavior of Companies

Regarding the research focused on entrepreneurial behavior, there are some studies related to the empirical analysis of various variables, where it is important to note that there is a significant relationship between the links of organizational capacity and entrepreneurial behavior (De Oliveira 2009; Svensson 2020). On the other hand, it is considered important that leadership is present for entrepreneurial behavior, since it is of utmost necessity when there is crisis and uncertainty; that is, to help cope with difficult environments (Horta and Kong 2014), and innovation is an element that contributes to improving economic growth (Wu and Huang 2008).

Similarly, emphasis is placed on studies about entrepreneurship from a human and social capital development perspective (Obschonka et al. 2012; Khoshmaram et al. 2020). Likewise, entrepreneurial behavior is not only empirically evidenced, but also addresses theoretical aspects, where it is necessary to deepen, understand, explain, and highlight the importance it has had over time and continues to have in organizations and society in general (Busenitz 2007; Gruber and Macmillan 2011; Teague and Gartner 2017). Entrepreneurial behavior has been considered a decisive element in organizations (Guachimbosa et al. 2019), as well as a key factor in generating strategies, providing various practical and theoretical benefits (Anderson et al. 2019), and also identifying relevant opportunities in the business context (Baltar and Brunet 2013; Andrés Pulgarín Molina and Acevedo 2011). Therefore, the importance of such behavior is drawing attention due to the organizational results it offers (Rutherford and Nagy 2014).

On the other hand, it is possible to highlight studies focused on the entrepreneurial behavior associating it with diverse variables, such as strategy, business success, and industrial environment, among others (Entrialgo et al. 2001). Similarly, entrepreneurial behavior within SMEs has also been analyzed in relation to other variables to clarify the link between them and behavior within organizations, such as sustainable development, where there is a positive association, indicating that while a company is focused on entrepreneurship it is likely to be committed to sustainable development (Iqbal and Malik 2019; Ayuso and Navarrete-Báez 2018).

### 1.3. Characteristics of the Business Manager

Several investigations have shown the relevant performance that organizations have had and the role they play within the economy; this is due to the characteristics of the owners, managers, and directors, such as demographic characteristics (Zhang 2017; Kellermanns et al. 2008; Peni 2014; Nguyen et al. 2018; Saidu 2019; Davidson et al. 2006; Yeoh and Hooy 2020; McKnight et al. 2000) which include seniority and educational level, among others. Other characteristics of a manager are those that are identified by the way they carry out their management; the way they establish their goals and objectives, both strategically and personally; how they make decisions; and how they delegate actions within the company (Mukhtar 2002). All of the above characteristics of managers have an impact on company performance, as they are considered management activities, such as entrepreneurship, perception of social responsibility, as well as social capital (Kim and Jung 2015).

On the other hand, there are other characteristics of managers that allow a better performance for the company, such as experience, choice of a successor, tenure (Newman et al. 2018), gender diversity, time in office, duality (Rubino et al. 2017), strategic change, knowledge, cultural aspects (Le and Kroll 2017), and business education, that in one way or another affect the results and performance of the company (Pascal et al. 2017). Within the research analyzed and focused on the characteristics of the manager, those oriented to personality and educational characteristics stand out, since they can have a

dominant influence on the members of the organization (Haas and Speckbacher 2017) and the ownership of the company (Yang et al. 2020); in addition to characteristics such as mental capacity, personality traits and ethics, among others, due to the importance they have both in companies and in society in general (Omri and Becuwe 2014).

In addition to the above, there are also characteristics catalogued as attributes of the managers that are considered to be of a subjective nature, such as the command of languages, degree of study, personality traits, and aggressiveness, among others, which directly impact the attitude of the members of the company and the dynamism of the work (Río Araújo 2006). On the other hand, other relevant characteristics of the manager in small and medium enterprises are processes and management functions, the results of the management process, components of the organizational environment (Foxley 1980), the way of working, decision making, aspects of ownership, and succession, among others, which have been considered factors of improvement with benefits and advantages to the organization (Shih and Wickramasekera 2011).

As can be observed, research works oriented to the entrepreneurial behavior in SMEs and the characteristics of their managers are scarce. One of them considers the characteristics of the owners/managers of the companies for their individual entrepreneurial behavior (Entrialgo et al. 1999), while another one accounts for the characteristics of the managers in the entrepreneurial behavior of the company (Suárez et al. 2000), that is, it goes from the individual to the organizational. It is precisely in this possibility of study that the following question arises as a research question: what is the difference in the entrepreneurial behavior of SMEs based on the demographic characteristics of their managers?

To achieve the objective of this research, which is to determine the difference in the entrepreneurial behavior of companies based on the demographic characteristics of their managers and to publicize compliance, a section on materials and methods is included, in which the subjects of study, the instrument used, and the statistical analysis used are described. Subsequently, the results show the main findings, in which the entrepreneurial behavior of companies varies in relation to the age and education of the managers; however, gender and seniority are not differentiating factors.

## 2. Materials and Methods

### 2.1. Sample and Procedure

This document is generated from a quantitative and transversal research study focused on SMEs in Ciudad Obregon, a city located in the northern state of Sonora, Mexico. The study was based on the agreement that establishes the stratification of companies in Mexico, which considers the size of the company in relation to the number of employees, ranging from 11 to 100 companies in the case of SMEs in the commercial and service sectors, and from 11 to 250 in industrial SMEs (Secretaría de Economía (México)). To determine the study population, the National Statistical Directory of Economic Units (DENUE) of the National Institute of Statistics and Geography (INEGI) was consulted (Instituto Nacional de Estadística y Geografía (México)), from which a convenience sample of 262 companies was selected, distributed as follows: 47 industrial, 89 service, and 126 commercial companies.

Visits were made to each of the establishments during the period from September 2018 to March 2019. The instrument was applied to the most senior manager in the organization, being mainly the owner or generally responsible for the organization (administrator, director, or manager). In some cases, it was necessary to make several visits to the same company in order to have the questionnaires answered by the managers of the company.

### 2.2. Measures and Instrument

An instrument with two sections (Ochoa 2019) was designed to collect the information. The first section includes general information on both the companies and the informants (managers), as shown in Table 1.

**Table 1.** Items and scales for general information of companies and managers.

| Item | Answer Options |
|---|---|
| **Company** | |
| Main Activity | Service, Trade, Industry |
| Type of Company | Family, Non- family |
| Seniority (Years) | Less than 10 years, 11 to 20 years, 21 to 30 years, 31 to 40 years, more than 40 years |
| Company Ownership | Sole owner, more than one owner, franchise, stock exchange listing |
| Level of Operations | Local, regional, national, international |
| **Manager** | |
| Position | Owner, responsible of the company |
| Gender | Female, Male |
| Age (Years) | 20 to 40, 41 to 60, more than 60 |
| Seniority (Years) | 0 to 10, 11 to 20, 21 to 30, 31 or more |
| Education Level | Elementary and secondary education, Bachelor's Degree, Graduate's Degree |

The second part consists of four questions that are included in the entrepreneurial behavior, which could be answered using a Likert scale of 7 options, from 1—very much in disagreement—to 7—very much in agreement (Kellermanns et al. 2008) (see Table 2).

**Table 2.** Entrepreneurial Behavior Items and Reliability.

| Item | Reliability |
|---|---|
| Over the past three years, our firm has pioneered the development of breakthrough innovations in its industry | 0.86 |
| Our firm has introduced many new products or services over the past three years | |
| Our firm has emphasized making major innovations in its products and services over the past three years | |
| Our firm has emphasized taking bold, wide-ranging actions in positioning itself and its products or services over the past three year | |

### 2.3. Data Analysis

The Kolmogorov-Smirnov normality test was first used to make statistical calculations of mean differences in order to identify which demographic characteristics are related to entrepreneurial behavior (Casado et al. 2020). The results for the four items and the variable were of a significance of 0.000, so the null hypothesis that establishes the distribution as normal is rejected; therefore, non-parametric tests were applied.

A bivariate analysis with the Mann-Whitney test was applied to identify the existence or lack of significant differences in the gender of the manager in relation to the entrepreneurial behavior of the company; likewise, the Kruskal-Wallis test was utilized to calculate the existence or lack of significant differences in the seniority in the company, and the age and level of education of the manager, each aspect independently, in relation to the entrepreneurial behavior of the company. All calculations were analyzed in the statistical software SPSS, version 26.0.

## 3. Results

The research findings are concentrated in two main sections: the first one is about the description of the studied companies and the other one focuses on the characteristics of the manager in relation to their entrepreneurial behavior.

### 3.1. Socio-Demographic data of the Companies

The studied companies were 262, most of which have been in the market for more than 20 years, so that only less than 1% of them are less than 10 years old (see Table 3).

**Table 3.** Seniority of the companies.

| Years in Business | Frequency | Percentage |
|---|---|---|
| 31 or more | 106 | 40.5 |
| Between 21 and 30 | 84 | 32.0 |
| Between 11 and 20 | 70 | 26.7 |
| Less than 10 | 2 | 0.8 |
| Total | 262 | 100 |

Regarding the ownership of the companies, half of them are sole proprietorships and only one of them is listed on the stock exchange (see Table 4).

**Table 4.** Company Ownership.

| | Frequency | Percentage |
|---|---|---|
| Sole owner | 132 | 50.4 |
| More than one owner | 118 | 45.0 |
| Franchise | 11 | 4.2 |
| Company listed on the stock exchange | 1 | 0.4 |
| Total | 262 | 100 |

The market in which they have participation is mostly concentrated at a regional level, and a little less than 10% of them have international operations (see Table 5).

**Table 5.** Level of Operations.

| | Frequency | Percentage |
|---|---|---|
| Local | 96 | 36.6 |
| Regional | 80 | 30.5 |
| National | 61 | 23.4 |
| International | 25 | 9.5 |
| Total | 262 | 100 |

### 3.2. The Entrepreneurial Behavior

Taking as a basis the responses scale that goes from disagreement to agreement, the reference average has a value of 4, so that the average of the entrepreneurial behavior is equal to 5.37. Consequently, it can be said that the studied companies behave in an entrepreneurial way since they have been pioneers in the development of innovations ($\bar{x}$ = 5.19); moreover, they have introduced many new products or services ($\bar{x}$ = 5.29); they have made important innovations in their products and services ($\bar{x}$ = 5.41); and, finally, they have taken bold and far-reaching actions to position themselves and their products or services ($\bar{x}$ = 5.58) (see Table 6).

**Table 6.** Mean and standard deviation values per variable and item.

| | Mean | Standard Deviation |
|---|---|---|
| Entrepreneurial Behavior | 5.37 | 1.75 |
| Item 1. Innovations Development | 5.19 | 2.13 |
| Item 2. Innovations Introduction | 5.29 | 2.16 |
| Item 3. Innovations Achievements | 5.41 | 2.07 |
| Item 4. Positioning Actions | 5.58 | 1.90 |

### 3.3. Entrepreneurial Behavior and Characteristics of Managers

The previous section visualizes the entrepreneurial behavior of the companies using the average. In this one, the results of the differences between this behavior are presented

based on the demographic characteristics of the managers, such as gender, labor seniority, age, and level of study.

### 3.3.1. Gender

Mann-Whitney's U-test resulted in average ranges for male directors of 134.48 and for female directors of 127.18, a slightly higher difference in the male category (see Table 7). The Mann-Whitney U statistic has a value of 7830.50; however, the *p* value is 0.439, greater than 0.05 (see Table 8), so it can be established that there is no difference in the distribution of entrepreneurial behavior among the gender categories; that is, such behavior in the studied SMEs is the same in both male and female directed enterprises.

**Table 7.** Average gender ranges.

|  | Gender | N | Average Range |
|---|---|---|---|
| Entrepreneur Behavior | Male | 155 | 134.48 |
|  | Female | 107 | 127.18 |
|  | Total | 262 |  |

**Table 8.** Mann-Whitney U test for Gender.

| Stadistics | Value |
|---|---|
| Mann-Whitney's U test | 7830.50 |
| Wilcoxon W | 13,608.50 |
| Standard error | 597.20 |
| Asymptotic (bilateral test) | 0.439 |

### 3.3.2. Seniority in the Company

According to Table 9, the average ranges derived from the Kruskal-Wallis test have higher values in the categories with fewer years of age, while the Kruskal-Wallis statistic has a value of 3217; however, the value of *p* is 0.359, greater than 0.05 (see Table 10), so it can be said that there is no difference in the distribution of entrepreneurial behavior among the seniority categories of managers.

**Table 9.** Average ranges of seniority.

|  | Seniority | N | Average Range |
|---|---|---|---|
| Entrepreneurial Behavior | 0 to 10 years | 154 | 131.28 |
|  | 11 to 20 years | 70 | 141.19 |
|  | 21 to 30 years | 31 | 116.53 |
|  | 31 years or more | 7 | 105.86 |
|  | Total | 262 |  |

**Table 10.** Kruskal-Wallis Seniority Test.

|  | Entrepreneurial Behavior |
|---|---|
| Kruskal-Wallis H | 3.217 |
| Degree of freedom | 3 |
| Asymptotic (bilateral test) | 0.359 |

### 3.3.3. Age

According to Table 11, the average ranges derived from the Kruskal-Wallis test have higher values in the categories with fewer years of age of the manager, while the Kruskal-Wallis statistic has a value of 6.835, with a value of *p* = 0.033, less than 0.05 (see Table 12), so it can be said that there is a difference in the distribution of entrepreneurial behavior among the age categories of managers.

**Table 11.** Average Age Ranges.

| | Age | N | Average Range |
|---|---|---|---|
| Entrepeneurial Behavior | 20–40 years | 125 | 139.83 |
| | 41–60 years | 117 | 129.13 |
| | 61 years or more | 20 | 93.30 |
| | Total | 262 | |

**Table 12.** Kruskal-Wallis Age Test.

| | Entrepreneurial Behavior |
|---|---|
| Kruskal-Wallis H | 6.835 |
| Degree of freedom | 2 |
| Asymptotic (bilateral test) | 0.033 |

In order to be more precise about this difference in behavior, a comparison was made by the age of the manager. The average range in the category of managers between 20 and 40 years old is 139.83, and in the category of 61 years old and more it is 93.30 (see Table 11), with a value of $p = 0.030$ (see Table 13). It can be stated that companies managed by people whose age fluctuates between 20 and 40 years old have greater entrepreneurial behavior than those whose leader is 61 years old or more; however, this is the only difference that can be statistically validated, since the other two comparisons have an adjusted value of $p > 0.05$.

**Table 13.** Age Pair Comparisons.

| Sample 1-Sample 2 | Test Statistic | Standard Deviation | Test Statistical Deviation | Sig. | Adjusted Sig. [a] |
|---|---|---|---|---|---|
| 61 years or more-41–60 years | 35.832 | 18.162 | 1.973 | 0.049 | 0.146 |
| 61 years or more-20–40 years | 46.528 | 18.077 | 2.574 | 0.010 | 0.030 |
| 41–60 years-20–40 years | 10.696 | 9.655 | 1.108 | 0.268 | 0.804 |

Each row tests the null hypothesis that the distributions in Sample 1 and Sample 2 are equal. Asymptotic meanings are displayed (bilateral tests). The significance level is 0.05. [a] Significance values have been adjusted by Bonferroni correction for several tests.

3.3.4. Education Level

According to Table 14, the average ranges derived from the Kruskal-Wallis test have higher values in the categories with a higher level of management education, while the Kruskal-Wallis statistic has a value of 14.359 with a value of $p = 0.001 < 0.05$ (see Table 15); consequently it can be said that there is a difference in the distribution of entrepreneurial behavior among the categories of educational level of managers.

**Table 14.** Average ranges of educational levels.

| | Age | N | Average Range |
|---|---|---|---|
| Entrepreneurial Behavior | Elementary and Secondary Education | 75 | 104.19 |
| | Bachelor's Degree | 147 | 140.54 |
| | Graduate's Degree | 40 | 149.48 |
| | Total | 262 | |

**Table 15.** Kruskal-Wallis Test for Educational Level.

|  | Entrepreneurial Behavior |
|---|---|
| Kruskal-Wallis H | 14.359 |
| Degree of Freedom | 2 |
| Asymptotic (bilateral test) | 0.001 |

In order to be more precise about this difference in behavior, a comparison was made by level of study of the manager. The average range in the elementary and secondary education level category is 104.19 and the bachelor's degree is 140.54 (See Table 14), with an adjusted value of $p = 0.002$ between both (see Table 16), so it can be said that companies run by people with a bachelor's degree have greater entrepreneurial behavior than those whose leader has elementary or secondary studies. When contrasting the elementary and secondary education category with an average range of 104.19 and those with graduate studies with a range of 149.48, with an adjusted value of $p = 0.006$ between both (see Table 16), it can be stated that companies managed by people with graduate studies have a higher entrepreneurial behavior than those whose manager has elementary or secondary education; however, when comparing the categories of bachelor's degree and graduate studies, it was found that there is no difference that can be statistically validated, since it has an adjusted value of $p > 0.05$.

**Table 16.** Educational Level Pair Comparisons.

| Sample 1-Sample 2 | Test Statistic | Standard Error | Test Statistical Deviation | Sig. | Adjusted Sig. [a] |
|---|---|---|---|---|---|
| Elementary and Secondary Education-Bachelor's Degree | −36.358 | 10.651 | −3.413 | 0.001 | 0.002 |
| Elementary and Secondary Education-Graduate's Degree | −45.288 | 14.696 | −3.082 | 0.002 | 0.006 |
| Bachelor's Degree-Graduate's Degree | −8.931 | 13.386 | −0.667 | 0.505 | 1.000 |

Each row tests the null hypothesis that the distributions in Sample 1 and Sample 2 are equal. Asymptotic significance (bilateral tests) is displayed. The significance level is 0.05. [a] Significance values have been adjusted by Bonferroni correction for several tests.

## 4. Discussion

Entrepreneurial behavior can be viewed from the perspective of the social aspect, that is, beyond for-profit entrepreneurship (Gruber and Macmillan 2011), as well as its application at the country level (Năstase and Kajanus 2010); nevertheless, the focus of this research has been entrepreneurial, studying the companies. In this sense, the analyzed Mexican companies exhibit entrepreneurial behavior, joining others that have been previously studied (Ayuso and Navarrete-Báez 2018). This can be seen in Indian SMEs that are related to their internationalization (Javalgi and Todd 2011); in England, which has an impact on the development of new products (Liu et al. 2017) and rural SMEs that show innovation (Blanchard 2017); in New Guinea in indigenous SMEs (Rante and Warokka 2013); in Pakistan (Iqbal and Malik 2019); and in Spain (Entrialgo et al. 2001; Ayuso and Navarrete-Báez 2018). The above can be interpreted as a limited number of studies on this topic in SMEs, but at the same time located in various environments around the world.

In addition, the most relevant findings generated from the tests can be grouped into two. The first group is related to the characteristics of the managers that do not present differences among them in the entrepreneurial behavior. In this case, gender can be mentioned, in such a way that there is no difference in the entrepreneurial behavior of the studied organizations when they are managed by men or women. This contrasts with studies in which women play an important role in family type organizations (Acheampong 2018), and on the difference between the ways organizations run by men and women are managed (Mukhtar 2002), especially due to cognitive and emotional issues (Li et al. 2019).



The seniority of the company's manager is not relevant for such behavior either; that is, the difference in the years that managers have been working is not a determinant for organizations to undertake different projects. This may be subject to debate since there is evidence that greater seniority at work has an influence on the knowledge of the job and the ability to perform it (Schmidt et al. 1986); while less experience leads to greater cultural intelligence (Puyod and Charoensukmongkol 2019). To keep the discussion open and unfinished, it was found that the most entrepreneurial companies have management teams with more previous joint experience, but at the same time with significant experience diversity (Suárez et al. 2000).

Contrary to the two previous aspects, a difference was found in the distribution of the entrepreneurial behavior of the companies in relation to the age of the managers, particularly those managed by people between 20 and 40 years old who have a greater entrepreneurial behavior than those managed by those who are 61 years old or older; nevertheless, the 41–60 age group showed no difference from either of the other two age groups. Without being conclusive, there is an approach in identifying age as a factor that may interfere in the entrepreneurial behavior. In that sense, there are previous studies in which the most entrepreneurial companies have management teams with a lower average age (Suárez et al. 2000); likewise, executive directors in the early stages of their careers tend to make more risky decisions, and after the age of 40, they begin to decline (Yeoh and Hooy 2020). One idea that can be considered socialized and confirmed by some studies is that young people are more entrepreneurial than older people (Lévesque and Minniti 2006), yet a positive self-image based on age increases the likelihood of being an entrepreneur regardless of chronological age (Kautonen et al. 2015).

The level of education of managers is another aspect in which there is a significant difference in the distribution of entrepreneurial behavior of companies. Specifically, companies that are managed by people with undergraduate or graduate studies show greater entrepreneurial behavior than those whose manager has non-professional studies (high school or less). Previous studies related to the above state that the most entrepreneurial companies have management teams with high technical training (Suárez et al. 2000), since the education of the CEO has positive effects on company indicators (Saidu 2019; Haas and Speckbacher 2017). Not only does having an education have better results, but also those with an entrepreneurial education perform significantly better than those with other educational backgrounds (Pascal et al. 2017).

In summary, it can be said that the entrepreneurial behavior of the companies shows differences based on the analysis of the age categories and the educational level of the managers. Seniority in the company and the gender of the person responsible for the company are not significant differentiating elements of this type of behavior, at least in the studied companies.

This work has important contributions to knowledge, especially in increasing the endless debate on the characteristics of those who lead companies and their impact on business performance. Although there is an approach and a contribution in that sense, it should be mentioned that the non-parametric statistical analysis limits statements of greater scope; in addition, the approach to SMEs delimits a more complex organizational reality such as the one existing in large companies. Finally, focusing on only four demographic characteristics of managers achieves a limited vision of the intangible reality in the management of companies in which qualities such as the way they communicate, negotiate, motivate, manage conflicts, and make decisions, among others, are left out.

The above limitations generate areas of opportunity and future study areas to expand this work, complement it, or move it to another level. It is proposed to carry out studies in which other study variables are correlated with the entrepreneurial behavior of companies or their performance; which involve the use of statistical tests with greater precision and depth that allow greater generalization; expand to other sizes, sectors and geographical regions that achieve a greater number of study subjects; and incorporate, in addition to

demographic characteristics, personal characteristics that allow the analysis of knowledge, skills, and attitudes of managers for the performance of their work.

The directors of the studied companies are characterized by being mostly men, with less than 20 years of service in the company, with bachelor's degrees, and approximately half of them are under 40 years old. Some of these characteristics can achieve a balance in the relationship between work and family; in the same way, they contribute to the entrepreneurial performance of the companies since family-to-work conflict (FWC) and family-to-work enrichment (FWE) shape the performance of entrepreneurial firms directly and indirectly (Lu et al. 2020). This demonstrates the opportunity for future studies, in which the direct relationship of age and seniority of managers with the work-family relationship is addressed in depth.

## 5. Conclusions

The demographic characteristics of the managers have a differentiating relationship in the behavior of the studied companies, especially with regard to age and educational level. This is not the case with gender and seniority, since they did not present significant differences. With this research, possibilities are generated for studies to be carried out in large companies, in other productive sectors or different types of companies and even non-profit sectors, as well as to include behavioral characteristics as study variables.

**Author Contributions:** S.O.J.: Formal analysis, funding acquisition, investigation, methodology, project administration, validation, writing—original draft & editing. A.R.G.G.: Data curation, software, formal analysis, investigation, methodology, validation, writing—original draft & editing. B.A.L.O.: Formal analysis, methodology, validation, writing—review & editing. S.V.d.R.: Methodology, validation, writing—original draft & editing. All authors have read and agreed to the published version of the manuscript.

**Funding:** This research was funded by Program for Strengthening Educational Excellence (PRO-FEXCE), Program for Promotion and Support of Research (PROFAPI) of the Sonora Institute of Technology (ITSON).

**Institutional Review Board Statement:** The study was conducted according to the guidelines of the Declaration of Helsinki and was approved by the Institutional Review Board of the Sonora Institute of Technology (File Number 2018-080, 31 March 2018).

**Informed Consent Statement:** Not applicable.

**Data Availability Statement:** The data presented in this study are available on request from the corresponding author.

**Conflicts of Interest:** The authors declare no conflict of interest

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
