# Peer review of "Entrepreneurial Behavior of SMEs and Characteristics of the Managers of Northwest Mexico"

_socsci, doi:10.3390/socsci10010008_

Round 1

Reviewer 1 Report

The paper deals with the study on „Entrepreneurial behavior of SMEs in relation to management characteristics in Northwest Mexico”. The presented topic is of high professional and practical interest what brings a significant added value to potential target group of readers. Hopefully, my remarks, observations and possible suggestions might bring the authors benefits for the enhancement of the paper to be published properly. Accordingly, I am stating my comments below.

Originality

From my point of view, the researched study can be considered as an innovative scientific contribution with clear identification of its overall structure.

Title

The title reflects the objective and content of the paper, the lenght is adequate.

Abstract

This part demonstrates the structured summary, background, results, conclusion, and impact of key findings.

Introduction

This part of the paper is properly designed in a correct explanatory way. The most relevant part is the introduction section that gives a perfect context for the justification of the research. This section is very well based and combined with Literature part.

I think references are almost to justify this topic. But it is critical to highlight what is the gap and what is new in this study. I would encourage the authors to provide more recent references (from the last 5 years). You need to develop this part. The manuscript could be even more sufficiently supported by evidence or proper references to work done elsewhere (Social Sciences Scopus and Web of Science listed journal).  I think references are not enough to justify this topic. For this purpose authors could downloads the following published article:

https://www.emerald.com/insight/content/doi/10.1108/IJPDLM-01-2017-0043/full/html?casa_token=K7CMx--SL0wAAAAA:gHBD2I_DlbqIdCS2oZvdccg_sH-FjD6uuLJZ3j44D4cjhVejMFEMeZ52U8tgn9mh5vYahfTRMvl8AQk4BUgZ6zK3aSdluyuuuqWy9IqDuU3r3isUIw

https://www.mdpi.com/2071-1050/11/7/1853

Materials and Method

Methodology is suitable for the research objectives.

Results

The research findings are concentrated in two main sections: the first one is about the description of the studied companies; and the other one focuses on the characteristics of the manager in relation to their entrepreneurial behavior.

Please add more results more analysis to this part. It is too week. The explanations are missing only tables.

Discussion

The discussion section should contain a sound discussion of the insights that can be gained from this work, a discussion of its limitations, and an outlook on future research opportunities.

Conclusions

In this section the main ideas of the manuscript are presented, the obtained results and their novelty are demonstrated. The interpretations and conclusions are sound and justified by the results.

Recommendation: major revision.

Author Response

Response to Reviewer 1 Comments

Point 1: Originality
From my point of view, the researched study can be considered as an innovative scientific contribution with clear identification of its overall structure.

Response 1: We appreciate your comments regarding the research.

Point 2: Title
The title reflects the objective and content of the paper, the lenght is adequate.

Response 2: We appreciate your comments regarding the research.

Point 3: Abstract
This part demonstrates the structured summary, background, results, conclusion, and impact of key findings.

Response 3: We appreciate your comments regarding the research.

Point 4: Introduction
This part of the paper is properly designed in a correct explanatory way. The most relevant part is the introduction section that gives a perfect context for the justification of the research. This section is very well based and combined with Literature part.

I think references are almost to justify this topic. But it is critical to highlight what is the gap and what is new in this study. I would encourage the authors to provide more recent references (from the last 5 years). You need to develop this part. The manuscript could be even more sufficiently supported by evidence or proper references to work done elsewhere (Social Sciences Scopus and Web of Science listed journal).  I think references are not enough to justify this topic. For this purpose authors could downloads the following published article:

https://www.emerald.com/insight/content/doi/10.1108/IJPDLM-01-2017-0043/full/html?casa_token=K7CMx--SL0wAAAAA:gHBD2I_DlbqIdCS2oZvdccg_sH-FjD6uuLJZ3j44D4cjhVejMFEMeZ52U8tgn9mh5vYahfTRMvl8AQk4BUgZ6zK3aSdluyuuuqWy9IqDuU3r3isUIw

https://www.mdpi.com/2071-1050/11/7/1853

Response 4: In the introduction section, the literature review was expanded, new consultations from the last five years were added with an impact factor, and existing consultations were analyzed in more detail. 18-169

Point 5: Materials and Method
Methodology is suitable for the research objectives.

Response 5: We appreciate your comments regarding the research.

Point 6: Results
The research findings are concentrated in two main sections: the first one is about the description of the studied companies; and the other one focuses on the characteristics of the manager in relation to their entrepreneurial behavior.

Please add more results more analysis to this part. It is too week. The explanations are missing only tables.

Response 6: The study approach was adjusted from introduction to conclusion. Other statistical calculations were added and the objective, method, results and conclusion were aligned.  242-355

Point 7: Discussion
The discussion section should contain a sound discussion of the insights that can be gained from this work, a discussion of its limitations, and an outlook on future research opportunities.

Response 7: In the discussion section, in addition to clarifying some issues arising from the results, limitations and suggestions for future studies were added.  358-424

Point 8: Conclusions
In this section the main ideas of the manuscript are presented, the obtained results and their novelty are demonstrated. The interpretations and conclusions are sound and justified by the results.

Response 8: We appreciate your comments regarding the research.

Reviewer 2 Report

The authors must get a native reader to read over the manuscript. The grammatical issues and sentence structures tend to be a distraction from the authors’ ideas.

The authors may want to explicitly establish a research question (s) or hypotheses to make the objectives of the study that much clearer.

The authors should introduce a literature review section and improve the literature. SME has generated considerable research over the past few decades. Thus, there is much available research that can be drawn upon to improve the manuscript.

Table 2 has 2 Cronbach’s alphas. I’m sure this is an error and should be corrected.

There is a mismatch between what the authors sought to achieve, the tests performed, and the conclusions that are presented

Line73: It is precisely in this possibility of study that this work of investigation arises, whose objective is to determine which demographic characteristics of the managers have greater interference in the enterprising behavior of the company, considering the previous results of the incipient studies.

Line 252: As it has been analyzed and reaffirmed with previous research, the younger the age and the higher the level of studies, the more related to the entrepreneurship of the companies.

However, the Kruskal-Wallis test that was conducted only shows for example that the distribution of entrepreneurial behavior differs among seniority categories. The tests do not show how they differ. It could very well be that the older the manager, the more entrepreneurial an organization is. In order to rule this out and to support the authors’ claims, additional tests should be conducted. Because the authors want to focus on descriptive statistics, the correlational analysis would provide some insight. Apart from significance, it would provide information on the direction of these relationships.

Best of luck!

Author Response

Response to Reviewer 2 Comments

Point 1: The authors must get a native reader to read over the manuscript. The grammatical issues and sentence structures tend to be a distraction from the authors’ ideas.

Response 1: An overall review of the text was conducted and editorial adjustments were made.

Point 2: The authors may want to explicitly establish a research question (s) or hypotheses to make the objectives of the study that much clearer.

Response 2: In the introduction section, a research question and objective were added, both provide greater focus to the research.  Lines: 156-169

Point 3: The authors should introduce a literature review section and improve the literature. SME has generated considerable research over the past few decades. Thus, there is much available research that can be drawn upon to improve the manuscript.

Response 3: In the introduction section, the literature review was expanded, new consultations from the last five years with an impact factor were added, and existing consultations were analyzed in greater detail. Lines: 15-155

Point 4: Table 2 has 2 Cronbach’s alphas. I’m sure this is an error and should be corrected.

Response 4: It was indeed a mistake; one of the values was removed.  Line: 201

Point 5:
There is a mismatch between what the authors sought to achieve, the tests performed, and the conclusions that are presented

Line73: It is precisely in this possibility of study that this work of investigation arises, whose objective is to determine which demographic characteristics of the managers have greater interference in the enterprising behavior of the company, considering the previous results of the incipient studies.

Line 252: As it has been analyzed and reaffirmed with previous research, the younger the age and the higher the level of studies, the more related to the entrepreneurship of the companies.

However, the Kruskal-Wallis test that was conducted only shows for example that the distribution of entrepreneurial behavior differs among seniority categories. The tests do not show how they differ. It could very well be that the older the manager, the more entrepreneurial an organization is. In order to rule this out and to support the authors’ claims, additional tests should be conducted. Because the authors want to focus on descriptive statistics, the correlational analysis would provide some insight. Apart from significance, it would provide information on the direction of these relationships.

Response 5: The study approach was adjusted from the summary to conclusion. Other statistical calculations were added and the objective, method, results and conclusion were aligned.  Line 164, Lines: 301-355 and Line 427.

Round 2

Reviewer 1 Report

The authors have revised their paper to address all my comments.

I am satisfied with their revision.

Author Response

The authors have revised their paper to address all my comments.

I am satisfied with their revision.

Thank you for your comments.